# Skip-Attention: Improving Vision Transformers by Paying Less Attention

**Shashanka Venkataramanan**[*†]
Qualcomm AI Research[‡]

**Amir Ghodrati**[†]
Qualcomm AI Research[‡]

**Yuki M. Asano**
University of Amsterdam

**Fatih Porikli**
Qualcomm AI Research[‡]

**Amirhossein Habibian**
Qualcomm AI Research[‡]

## ABSTRACT

This work aims to improve the efficiency of vision transformers (ViTs). While ViTs use computationally expensive self-attention operations in every layer, we identify that these operations are highly correlated across layers – a key redundancy that causes unnecessary computations. Based on this observation, we propose SKIPAT, a method to reuse self-attention computation from preceding layers to approximate attention at one or more subsequent layers. To ensure that reusing self-attention blocks across layers does not degrade the performance, we introduce a simple parametric function, which outperforms the baseline transformer's performance while running computationally faster. We show that SKIPAT is agnostic to transformer architecture and is effective in image classification, semantic segmentation, image denoising, and video denoising. We achieve improved throughput at the same-or-higher accuracy levels in all these tasks. Code can be found at https://github.com/Qualcomm-AI-research/skip-attention

## 1 Introduction

The transformer architecture (Vaswani et al., 2017) has become an important and highly influential model family, due to its simplicity, scalability, and its wide range of applications. While originally stemming from the domain of natural language processing (NLP), with the advent of the Vision transformer (ViT) (Dosovitskiy et al., 2020), this has become a standard architecture in computer vision, setting various state-of-the-art (SoTA) performances on tasks ranging from representation learning, semantic segmentation, object detection and video understanding (Caron et al., 2021; Liu et al., 2021; Carion et al., 2020; Liang et al., 2022; Girdhar et al., 2019).

However, the original formulation of the transformer includes a quadratic computational complexity with respect to the number of input tokens. Given that this number typically ranges from $14^2$ for image classification all the way to $128^2 = 16K$ for image denoising, this constraint on memory and compute severely limits its applicability. To tackle this problem, there have been three sets of approaches. The first leverages redundancies across input tokens and simply reduces computation by efficient sampling, *e.g.*, dropping or merging redundant tokens (Tang et al., 2022; Fayyaz et al., 2022; Yin et al., 2022). This, however, means that the final output of the ViT is not spatially continuous and can thus not be used beyond image-level applications such as semantic segmentation or object localization. The second set of approaches aims to cheaply estimate the attention computation, but generally at the cost of reduced performances (Yu et al., 2022; Chu et al., 2021).

In this work, we propose a novel, so far unexplored approach to solving this problem: simply approximating the computationally expensive blocks of the transformer with a much faster, simpler parametric function. To arrive at this solution, we first thoroughly analyse the crucial multi-head self-attention (MSA) block of the ViT. Through this analysis, we find that the attention of the CLS

---

[*]Work done during internship at Qualcomm AI Research

[†]equal contribution

[‡]Qualcomm AI Research is an initiative of Qualcomm Technologies, Inc

Figure 1: **Performance of SKIPAT across 5 different tasks.** Our novel SKIPAT method achieves superior accuracy *vs.* efficiency trade-off over the baseline transformer on a wide array of tasks. Circle areas are proportional to parameter count.

tokens to the spatial patches has a very high correlation across the transformer's blocks, thus leading to unnecessary computations. This motivates our approach to leverage attention from an early part of the model and simply reuse it for deeper blocks – basically "skipping" subsequent SA calculations instead of recomputing them at every layer.

Based on this, we go one step further and explore if the *entire* MSA block of a layer can be skipped by reusing the representation from previous layers. We find that a simple parametric function inspired from ResneXt's depth-wise convolutions (Xie et al., 2017) can outperform the baseline performance – while being computationally faster in terms of throughput and FLOPs. Previous works that use convolutions for improving efficiency in transformers have proposed *merging* convolution layers with transformer blocks (Graham et al., 2021), *refining* MSA representations by introducing convolutions inside MSA blocks (Zhou et al., 2021a;b), or *replacing* MSA blocks with MLP layers (Pan et al., 2022c). In contrast, we propose to leverage redundancies across MSA blocks and *approximate* them wholly using parametric functions. SKIPAT is general-purpose and can be applied to a ViT in any context: Figure 1 shows that our novel parametric function achieves superior accuracy *vs.* efficiency trade-off compared to the baseline transformer on a wide variety of tasks, datasets, and model sizes. SKIPAT is architecture agnostic and can be applied to isotropic, hierarchical, and hybrid transformer architectures resulting in superior performances than the baseline.

In summary, our main contributions are as follows:

1. We propose a novel plug-in module that can be placed in any ViT architecture for reducing the costly $\mathcal{O}(n^2)$ Self-Attention computations
2. We show that SKIPAT is agnostic to transformer architecture and achieves state-of-the-art performances in throughput at same-or-better accuracies for ImageNet, Pascal-VOC2012, SIDD, DAVIS and ADE20K (in the latter of which we obtain 40% speedup)
3. We further demonstrate the generality of our method by obtaining a 26% reduction in self-supervised pretraining time (at no downstream accuracy loss) and by demonstrating superior on-device latency
4. Finally, we analyse the sources of performance gains and extensively ablate our method to provide a model family which can be used for trading off accuracy and throughput

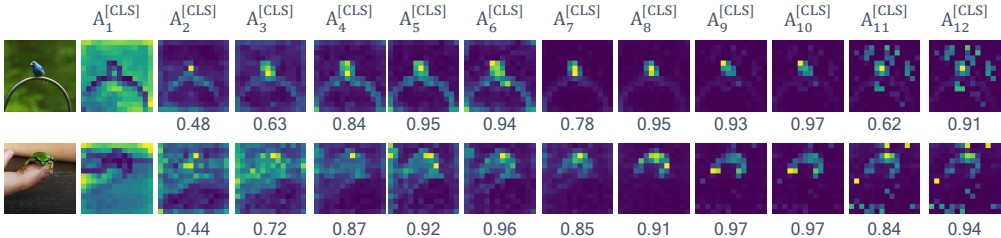

Figure 2: *Attention correlation*. Mean of the attention heads from the CLS token of a pretrained ViT-T/16 at different layers from the validation set of ImageNet-1K. Numbers below each attention map indicates the cosine similarity of $A_l^{\text{[CLS]}}$ with $A_{l-1}^{\text{[CLS]}}$.

## 2 RELATED WORK

Great effort has been made to improve the efficiency of vision transformers (ViT) (Dosovitskiy et al., 2020) from multiple aspects:

**Token sampling** improves the efficiency either by restructuring images during the tokenization step (Yuan et al., 2021; Han et al., 2021), pruning the redundant tokens over training (Kong et al., 2022; Tang et al., 2022) or dynamically at inference (Yin et al., 2022; Rao et al., 2021; Fayyaz et al., 2022; Chen et al., 2021). Despite their effectiveness in reducing the computational cost in image classification, token sampling methods are hardly applicable to dense prediction tasks, *e.g.* semantic segmentation and image denoising, where the output image should be spatially continuous. Our approach is complementary to these methods and achieves on-par or higher performance on both classification and dense prediction tasks.

**Hybrid architectures** such as Uniformer (Li et al., 2022), MobileViT (Mehta & Rastegari, 2021), and others (Liu et al., 2022; Pan et al., 2022a; Mehta & Rastegari, 2022), incorporate efficient convolutional modules into vision transformers. These architectures achieve this by employing MobileNet blocks in Uniformer, MobileNetV2 blocks in MobileViT, or using stacks of convolutions in the image tokenization step (Graham et al., 2021; Wu et al., 2021). Other approaches disentangle high and low-frequency representations in MSA blocks (Pan et al., 2022b) or replace MSA

| How do they improve efficiency?
Does the method satisfy each property? | Token
sampling | Hybrid
network | Efficient
attention | SKIPAT |
|---|:---:|:---:|:---:|:---:|
| Improve *throughput* during inference? | ✓ | ✓ | ✗ | ✓ |
| Generalize to *dense prediction tasks*? | ✗ | ✓ | ✓ | ✓ |
| Tackle *quadratic complexity* of self-attention? | ✗ | ✗ | ✓ | ✓ |
| Generalize to *different* transformer backbones? | ✗ | ✗ | ✗ | ✓ |

Table 1: SKIPAT vs. *vision transformers.* Comparison between SKIPAT and methods that improve the efficiency of vision transformers. Among the listed methods, only SKIPAT satisfies all the listed properties.

blocks in the early layers of the network with MLP layers (Pan et al., 2022c). In our work, we use convolutions to accelerate the computation of vision transformers. However, instead of crafting custom blocks, as done in (Mehta & Rastegari, 2021; Pan et al., 2022a; Mehta & Rastegari, 2022; Li et al., 2022; Pan et al., 2022c), we *approximate* entire MSA block using convolutions. This enables us to apply our method across isotropic, hierarchical, and hybrid transformer architectures. We compare SKIPAT with the existing methods relevant to improving the efficiency of vision transformers in Table 1 and show that among the listed methods, only SKIPAT shows all the listed properties.

**Efficient attention** methods aim to reduce the quadratic complexity of the self-attention operation in vision transformers. Several approaches have been proposed, such as global downsampling of key and value embeddings (Wang et al., 2021a; 2022a; Wu et al., 2021), performing self-attention in local windows (Liu et al., 2021), alternating between local and global self-attentions (Chu et al., 2021; Mehta & Rastegari, 2021; Pan et al., 2022a), or replacing self-attention with a simple pooling (Yu et al., 2022). However, reducing self-attention to a local neighborhood limits their ability to model long-range dependencies, leading to significant performance degradation with only moderate speedups (Zhang et al., 2021). In addition, some methods, such as cyclic shift in Swin (Liu et al., 2021), lack efficient support, thus reducing actual efficiency gains in terms of latency. In contrast, our method relies on a few blocks with strong, yet inefficient self-attention operators and lighter, accurate attention estimators in other blocks. As the estimators only use standard convolutional operations, our method translates to actual latency gains. The approach of using convolution layers is similar to (Zhou et al., 2021b;a), that introduce convolutions inside MSA to *refine* attention maps. However, adding a convolution operation at every layer increases computation overhead. Additionally, (Xiao et al., 2019; Wang et al., 2021b; Ying et al., 2021) observed redundancies in attention maps for NLP tasks. Instead of copying attention maps, we propose an efficient parametric function that achieves high throughput while maintaining high model performance in vision tasks.

## 3 SKIP-ATTENTION

### 3.1 PRELIMINARIES

**Vision Transformer.** Let $x \in \mathbb{R}^{h \times w \times c}$ be an input image, where $h \times w$ is the spatial resolution and $c$ is the number of channels. The image is first tokenized into $n = hw/p^2$ non-overlapping patches, where $p \times p$ is patch size. Each patch is projected into an embedding $z_i \in \mathbb{R}^d$ using a linear layer to obtain the tokenized image:

$$Z_0 = (z_1; \dots; z_n) \in \mathbb{R}^{n \times d} \qquad (1)$$

Here, "; " denotes row-wise stacking. Positional embeddings are added to $Z_0$ to retain positional information. The token embeddings are then input to a $\mathcal{L} = \{1, \ldots, L\}$ layer transformer whose output is denoted as $Z_L$. In the supervised setting, a learnable token $z^{[\text{CLS}]} \in \mathbb{R}^d$ is prepended to the tokenized image in (1) as $Z_0 := (z^{[\text{CLS}]}; Z_0) \in \mathbb{R}^{(n+1) \times d}$.

**Transformer Layer.** Every layer of the transformer consists of a multi-head self attention (MSA) block followed by a multi-layer perceptron (MLP) block. In the MSA block, the input, $Z_{l-1} \in \mathbb{R}^{n \times d}$, for $l \in \mathcal{L}$, is first projected into three learnable embeddings $\{Q, K, V\} \in \mathbb{R}^{n \times d}$. The attention matrix $A$, is calculated as

$$A := \sigma\left(\frac{QK^T}{\sqrt{d}}\right) \in \mathbb{R}^{n \times n} \tag{2}$$

where $\sigma(.)$ denotes the row-wise softmax operation. The "multi-head" in MSA is defined by considering $h$ attention heads where each head is a sequence of $n \times \frac{d}{h}$ matrix. The attention heads are reprojected back to $n \times d$ using a linear layer which is combined with the value matrix as

$$Z^{\text{MSA}} := AV \in \mathbb{R}^{n \times d} \tag{3}$$

The output representations from the MSA block is then input to the MLP block which comprises two linear layers separated by a GeLU activation (Hendrycks & Gimpel, 2016). At a given layer $l$, the computational flow of representations through a transformer block is denoted as

$$Z_l \leftarrow Z_l^{\text{MSA}} + Z_{l-1}, \tag{4}$$
$$Z_l \leftarrow \text{MLP}(Z_l) + Z_l. \tag{5}$$

Both the MSA and MLP blocks have residual connections with layer normalization (LN) (Ba et al., 2016). While MSA blocks in each layer of the transformer learn representations independently, in the next subsection, we show that empirically there exist high correlation across these layers.

## 3.2 MOTIVATION: LAYER CORRELATION ANALYSIS

**Attention-map correlation.** The MSA block in ViT encodes the similarity of each patch to every other patch as an $n \times n$ attention matrix. This operator is computationally expensive with $\mathcal{O}(n^2)$ complexity (2). As ViTs scale up, *i.e.*, as $n$ increases, the complexity grows quadratically and this operation becomes a bottleneck. Recent NLP works (Vig & Belinkov, 2019; Vig, 2019) have shown that self-attention across adjacent layers in SoTA language models exhibit very high correlation. This raises the question – *is it worth to compute self-attention at every layer of a vision transformer?*

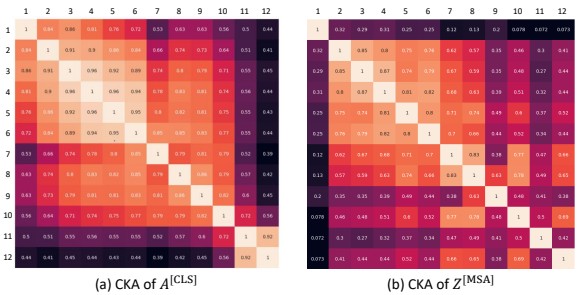

(a) CKA of $A^{[\text{CLS}]}$      (b) CKA of $Z^{[\text{MSA}]}$

Figure 3: *CKA analysis of $A^{[\text{CLS}]}$ and $Z^{\text{MSA}}$ across differ-ent layers of pretrained ViT-T/16. Vanilla ViT-T/16 has high correlation across both attention maps (layer 3 to 10) and $Z^{\text{MSA}}$ (layer 2 to 8)*

To address this question, we analyze the correlation of the self-attention maps across different layers of ViT. As shown in Figure 2, the self-attention maps from the class token, $A^{[\text{CLS}]}$, exhibit high correlation especially in the intermediate layers. The cosine similarity between $A_{l-1}^{[\text{CLS}]}$ and $A_l^{[\text{CLS}]}$ can be as high as 0.97, as indicated in the bottom of each attention map in Figure 2. We observe similar behavior from other token embeddings, which we analyze in the supplementary material. To quantify this correlation, we compute the Centered Kernel Alignment (CKA) (Kornblith et al., 2019; Cortes et al., 2012) between $A_i^{[\text{CLS}]}$ and $A_j^{[\text{CLS}]}$ for every $i, j \in \mathcal{L}$ across all validation samples of ImageNet-1K. CKA measures the similarity between representations obtained from intermediate layers of the network, where a high value of CKA indicates high correlation between the representations. From Figure 3 (a), we observe that ViT-T has a high correlation across $A^{[\text{CLS}]}$, especially from layers 3 through 10.

Figure 4: SKIPAT *framework* We illustrate SKIPAT on ViT (Dosovitskiy et al., 2020). The SKIPAT parametric function ($\Phi$) uses representations of the MSA block (in solid color) $Z_{l-1}^{\text{MSA}}$ as input, which undergoes a series of transformations. An element-wise summation ($\bigoplus$) with the output of the MLP block from layer $l-1$ and $\hat{Z}_l^{\text{MSA}}$ is used as input to the MLP block at layer $l$. The MSA operation (crossed out) is thus not computed and is discarded from the computational graph. With SKIPAT the total number of layers remains unchanged.

**Feature correlation.** In ViTs, the high correlation is not just limited to $A^{[\text{CLS}]}$, but the representation from MSA blocks, $Z^{\text{MSA}}$, also show high correlation throughout the model (Raghu et al., 2022). To analyze the similarity across these representations, we compute the CKA between $Z_i^{\text{MSA}}$ and $Z_j^{\text{MSA}}$ for every $i, j \in \mathcal{L}$. We observe from Figure 3 (b), that $Z^{\text{MSA}}$ also have high similarity across adjacent layers of the model especially in the earlier layers, *i.e.*, from layer 2 through 8.

### 3.3    IMPROVING EFFICIENCY BY SKIPPING ATTENTION

Based on our observation of high representation similarity across MSA blocks of a transformer (subsection 3.2), we propose to leverage the correlation across both the attention matrix and the representations from the MSA block to improve the efficiency of vision transformers. Instead of computing the MSA operation (3) independently at every layer, we explore a simple and effective strategy to utilize dependencies across the features from these layers.

In particular, we propose to skip MSA computation in one or more layers of a transformer by reusing representations from its adjacent layers. We term this operation as *Skip Attention* or SKIPAT. As the compute and memory benefit from skipping the entire MSA block is greater than skipping just the self-attention operation ($\mathcal{O}(n^2 d + n d^2)$ *vs.* $\mathcal{O}(n^2 d)$), in this paper we focus on former. However, instead of directly reusing features, *i.e.*, copying the features from the source MSA block to one or more adjacent MSA blocks, we introduce a parametric function. The parametric function ensures that directly reusing features does not affect the translation invariance and equivariance in these MSA blocks and acts as a strong regularizer to improve model generalization.

**SKIPAT parametric function**    Let $\Phi : \mathbb{R}^{n \times d} \to \mathbb{R}^{n \times d}$ denote the parametric function that maps output of the MSA block from $l-1$ to $l$ as $\hat{Z}_l^{\text{MSA}} := \Phi(Z_{l-1}^{\text{MSA}})$. Here, $\hat{Z}_l^{\text{MSA}}$ is the approximation of $Z_l^{\text{MSA}}$. The parametric function can be as simple as an identity function, where $Z_{l-1}^{\text{MSA}}$ is directly reused. Instead of computing MSA operation at $l$, we use $Z_{l-1}^{\text{MSA}}$ as the input to the MLP block at $l$. When using an identity function, due to the absence of MSA operation at $l$, the relation across tokens is no longer encoded in the attention matrix, which affects representation learning. To mitigate this, we introduce the SKIPAT parametric function inspired from ResNeXt (Xie et al., 2017) as shown in Figure 4, to encode local relations among tokens. The SKIPAT parametric function consists of two linear layers and a depth-wise convolution (DwC) (Chollet, 2017) in between, as follows:

$$\hat{Z}_l^{\text{MSA}} := \text{ECA}\Big(\text{FC}_2\Big(\text{DwC}\big(\text{FC}_1(Z_{l-1}^{\text{MSA}})\big)\Big)\Big) \qquad (6)$$

In the case of supervised learning, we first separate the CLS embeddings from $Z^{\text{MSA}} \in \mathbb{R}^{(n+1) \times d}$ into class embeddings $Z_C^{\text{MSA}} \in \mathbb{R}^d$ and the patch embeddings to $Z_P^{\text{MSA}} \in \mathbb{R}^{n \times d}$. The patch embeddings are then input to the first linear layer $\text{FC}_1 : \mathbb{R}^{n \times d} \to \mathbb{R}^{n \times 2d}$, which expands the channel dimension. This is followed by $\text{DwC} : \mathbb{R}^{\sqrt{n} \times \sqrt{n} \times 2d} \to \mathbb{R}^{\sqrt{n} \times \sqrt{n} \times 2d}$ with kernel $r \times r$. Note that before the DwC operation, we spatially reshape the input matrix to a feature tensor. Han *et al.* (Han et al., 2022) shows that the behavior of depth-wise convolution operation resembles local attention, which helps learn translation equivalent representations and also reduces the complexity of the parametric function. The output of the DwC is then flattened back to a vector and fed to the last FC layer

$FC_2 : \mathbb{R}^{n \times 2d} \to \mathbb{R}^{n \times d}$ which reduces the channel dimension back to its initial dimension $d$. We use GeLU activations after $FC_1$ and DwC. Following (Wang et al., 2020), we use efficient channel attention module (ECA) after $FC_2$ to enhance the cross-channel dependencies. The ECA module first aggregates the features along the channel dimension using global average pooling (GAP). A $1 \times 1$ convolution with adaptive kernel size proportional to channel dimension is applied followed by sigmoid activation. This operation of the ECA module enhances cross-channel dependencies. We then concatenate the embedding of the class-token with the output of the ECA to obtain $\hat{Z}_l^{\text{MSA}}$.

**SKIPAT framework.** The overall framework of SKIPAT is illustrated in Figure 4. SKIPAT can be incorporated into any transformer architecture which we empirically show in section 4. Depending on the architecture, one can skip the MSA operation in one or more layers of the transformer. In ViT, as we empirically observe that representations from the MSA block, $Z^{\text{MSA}}$, have high correlations from layer 2 through 7 (subsection 3.2), we employ the SKIPAT parametric function in these layers. This means that we use the $Z_2^{\text{MSA}}$ as input to the SKIPAT parametric function and skip MSA operations in layers 3-8. Instead, the features from the output of the SKIPAT parametric function is used as input to the MLP block. The computation flow of representations is now:

$$Z_l \leftarrow \Phi(Z_{l-1}^{\text{MSA}}) + Z_{l-1} \tag{7}$$

$$Z_l \leftarrow \text{MLP}(Z_l) + Z_l \tag{8}$$

Due to the presence of residual connections in the MSA and MLP blocks, which is standard in ViT (Dosovitskiy et al., 2020), the MLP blocks at layer 3 through 8 learn representations independently and cannot be discarded from the computational graph. It is important to note that, with SKIPAT the total number of layers in ViT remain unchanged, but there are fewer MSA blocks.

**Complexity: MSA *vs.* SKIPAT** The self-attention operation involves three operations. Firstly, the token embeddings are projected into query, key and value embeddings, secondly, attention matrix $A$ is computed as dot product between $Q$ and $K$ and finally, the output representations are computed as dot product between $A$ and $V$. This results in a complexity of $\mathcal{O}(4nd^2 + n^2d)$. Since $d \ll n$, the complexity of MSA block can be reduced to $\mathcal{O}(n^2d)$.

The SKIPAT parametric function consists of two linear layers and one depth-wise convolution, which results in a $\mathcal{O}(2nd^2 + r^2nd)$ complexity, where $r \times r$ is the kernel size of the DwC operation. The overall complexity of SKIPAT can be reduced to $\mathcal{O}(nd^2)$ since $r^2 \ll d$. Thus, SKIPAT has fewer FLOPs than MSA block as $\mathcal{O}(nd^2) < \mathcal{O}(n^2d)$ when $n$ increases as transformers scale up.

## 4 EXPERIMENTS

### 4.1 COMPARISON WITH STATE-OF-THE-ART

**Image Classification** We use isotropic transformer architectures like ViT-T/16 (Dosovitskiy et al., 2020), ViT-S/16 (Dosovitskiy et al., 2020), ViT-B/16 (Dosovitskiy et al., 2020), hierarchical architectures like PvT-T (Wang et al., 2021a), PvT-S (Wang et al., 2021a) and hybrid architectures like LIT-T (Pan et al., 2022c) and LIT-S (Pan et al., 2022c) as our backbone on ImageNet-1K. For fair comparisons, we follow the experimental settings in (Touvron et al., 2021), (Wang et al., 2021a) and (Pan et al., 2022c) to train ViT, PvT and LIT respectively. For ViT, we evaluate SKIPAT against SoTA methods: A-ViT (Yin et al., 2022), ATS (Fayyaz et al., 2022), PS-ViT (Tang et al., 2022), and Rev-Vit (Mangalam et al., 2022). To the best of our knowledge, these are all the works that improve the efficiency of ViT without modifying its underlying architecture.

From Table 2a, we observe that SKIPAT achieves the best accuracy *vs.* efficiency trade-off compared to all SoTA methods on different transformer backbones. Notably, we outperform different variants of ViT by 0.1% to 0.4% and improve throughput by 19%, to 25% . Interestingly, SoTA methods achieve lower accuracy or are on-par with the baseline. Since SKIPAT uses a parametric function to skip computing MSA blocks, our reduction in number of parameters and in FLOPs is comparable to the SoTA. Dehghani *et al.* (Dehghani et al., 2022) highlight the significance of using *throughput* as a metric to measure model efficiency: as the reduction in FLOPs does not necessarily correspond to improvements in latency, as it does not take into account the degree of parallelism or other hardware details. In line with this argument, we observe that while SoTA methods such as ATS (Fayyaz et al., 2022) achieve large reduction in FLOPs, they have lower throughput when compared to SKIPAT.

| Backbone | Method | top-1↑ (%) | Param↓ (×10^6) | GFlops↓ | Throughput↑ (im/s ×10^3) |
|---|---|---|---|---|---|
| ViT-T/16 | ViT | 72.8 | 5.7 | 1.2 | 5.8 |
| | A-ViT | 71.0 | 5.7 | 0.8 | 6.3 |
| | ATS | 72.7 | 5.7 | 0.9 | 6.1 |
| | PS-ViT | 72.6 | – | **0.7** | 6.6 |
| | SkipAT | **73.3** | 5.8 | 1.1 | 6.9 |
| ViT-S/16 | ViT | 79.8 | 22.4 | 4.6 | 3.2 |
| | A-ViT | 78.6 | 22.4 | 3.6 | 3.4 |
| | ATS | 79.7 | 22.4 | 2.9 | 3.3 |
| | PS-ViT | 79.4 | – | 2.6 | 3.9 |
| | Rev-ViT | 79.8 | 22.4 | 4.6 | 3.6 |
| | SkipAT | **80.0** | 22.1 | 4.0 | 3.8 |
| ViT-B/16 | ViT | 81.8 | 87.3 | 17.6 | 1.2 |
| | SViTE | 81.6 | **52.0** | 11.5 | 1.3 |
| | Rev-ViT | 81.5 | 87.3 | 17.6 | 1.2 |
| | PS-ViT | 81.5 | – | **9.8** | 1.6 |
| | SkipAT | **82.2** | 86.7 | 15.2 | 1.5 |
| PvT-T | PvT-T | 75.1 | 13 | 1.9 | 1.5 |
| | SkipAT | **76.1** | **12** | **1.7** | **1.8** |
| PvT-S | PvT-S | 79.8 | 25 | 3.8 | 1.3 |
| | SkipAT | **80.1** | **23** | **3.4** | **1.6** |
| LIT-T | LIT-v2-S | 82.0 | 28 | 3.7 | 1.4 |
| | LIT-T | 81.1 | 19 | 3.6 | 1.3 |
| | SkipAT | **81.4** | **18** | **3.4** | **1.4** |
| LIT-S | LIT-S | 81.5 | 27 | 4.1 | 1.3 |
| | SkipAT | **82.0** | **25** | **3.6** | **1.5** |

(a) *Image classification on ImageNet-1K.*

| Method | 224 × 224 | 384 × 384 |
|---|---|---|
| ViT-T/16 | 5.65 | 20.49 |
| ViT-T/16 + SkipAT | **4.76** | **15.22** |

(b) *On-device latency (msec)*

| Method | Jaccard↑ | CorLoc↑ |
|---|---|---|
| ViT-T | 32.2 | 39.5 |
| ViT-T + SkipAT | **38.0** | **41.5** |
| ViT-S | 29.0 | 40.6 |
| ViT-S + SkipAT | **34.0** | **41.2** |
| ViT-B | 33.6 | 36.4 |
| ViT-B + SkipAT | **36.8** | **37.2** |

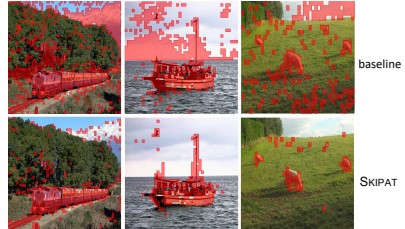

(c) *Unsupervised object discovery*

Table 2: (a) *Accuracy* vs. *efficiency* comparison of SkipAT with SoTA methods for image resolution 224 × 224. For all the methods, we measure throughput (image/sec) with a batch size of 1024 on a single NVIDIA A100 GPU, averaged over the validation set of ImageNet-1K. Additional comparisons are give in Table 7. (b) *On-device latency* of vanilla ViT *vs.* SkipAT for different image resolutions on a Samsung Galaxy S22 powered by Qualcomm Snapdragon 8 Gen 1. (c) *Unsupervised object discovery* using Jaccard similarity and Correct Localization (CorLoc), on the validation set of Pascal VOC2012. Image sources (from left to right): valley train (licensed under CC BY-SA 4.0), fishing boat (licensed under CC BY-SA 4.0), near Snowshill (licensed under CC BY-SA 4.0)

We also observe from Table 2a that SkipAT improves the performance of *pyramid architectures* PvT-T by 1.0% and improves throughput by 19%. On average, SkipAT outperforms variants of PvT with 20% gain in throughput. We also observe that SkipAT enhances the performance of *hybrid architectures* LIT with an average gain of 12% in throughput. Additionally, LIT-S + SkipAT achieves the same accuracy as baseline LIT-v2-S but with fewer parameters, FLOPs, and 7% gain in throughput. Thus, we show the ability of SkipAT to generalize to different transformer backbones.

**Visualizing attention maps and $Z^{\text{MSA}}$ correlation.** We analyze the effect of the SkipAT parametric function by visualizing the mean of attention heads of the CLS token from the last four layers of ViT-T/16. From Figure 5a, we observe that while attention maps from vanilla ViT (last two layers) do not solely attend to the object, the attention maps from SkipAT accurately focuses on the object. It is interesting to note that, the attention maps from SkipAT are also capable of attending to multiple objects in the image (Figure 5a: second example). The CKA of the representations from the MSA block in Figure 5b, shows that $Z^{\text{MSA}}$ has lower correlation across layers except between the layers where the MSA operation is skipped (layer 3 to 8). However, unlike vanilla ViT (Figure 3 (b)) the correlation from each layer to every other layer is quite low. This shows that our SkipAT parametric function acts as a strong regularizer and thus improves the representations of the model.

**Unsupervised object discovery.** We further analyze whether pretrained ViTs can attend to semantically meaningful regions of the image when evaluated on a different dataset without fine-tuning it. We follow (Caron et al., 2021), and visualize the segmentation masks produced from the final layer of the pretrained SkipAT on the Pascal-VOC12 (Everingham et al.). From Table 2(c),we observe that while vanilla ViT-S/16 does not accurately attend to the object, SkipAT is able to localize objects quite accurately without any fine-tuning. To quantify this observation, using Jaccard similarity

| METHOD | BACKBONE | mIOU↑ | GFLOPs↓ | THROUGHPUT↑ |
|---|---|---|---|---|
| | ResNet-101 (Yu et al., 2022) | 40.7 | 261 | 24.1 |
| Semantic FPN (Kirillov et al., 2019) | PoolFormer-S36 (Yu et al., 2022) | 42.0 | 191 | 8.4 |
| | PoolFormer-M36 (Yu et al., 2022) | 42.4 | 271 | 5.4 |
| | ResNet-18 (He et al., 2016) | 39.9 | 886 | 17.1 |
| | ResNet-101 (He et al., 2016) | 44.9 | 1031 | 12.0 |
| | Swin-T (Liu et al., 2021) | 45.8 | 945 | 14.2 |
| | ConvNeXt-T (Liu et al., 2022) | 46.7 | 939 | 15.7 |
| UperNet (Xiao et al., 2018) | ViT-T (Dosovitskiy et al., 2020) | 37.3 | 212 | 24.1 |
| | ViT-T + SKIPAT | **40.6** | **173** | **34.7** |
| | ViT-S (Dosovitskiy et al., 2020) | 44.4 | 360 | 19.5 |
| | ViT-S + SKIPAT | **45.3** | **283** | **27.2** |
| | ViT-B (Dosovitskiy et al., 2020) | 45.6 | 787 | 11.1 |
| | ViT-B + SKIPAT | **46.3** | **633** | **15.5** |

Table 3: *Semantic Segmentation on ADE20K.* All models are pretrained on ImageNet-1k and fine-tuned on ADE20K. Following Swin (Liu et al., 2021) and ConvNeXt (Liu et al., 2022), we report mIoU with multi-scale testing. FLOPs and throughput are calculated on the input size of $2048 \times 512$. Throughput of all models are measured with a batch size of 1 on a single NVIDIA A100 GPU, averaged over 100 forward passes.

and CorLoc (Melas-Kyriazi et al., 2022). As shown in Table 2(c), SKIPAT outperforms different variants of vanilla ViT with a significant gap in terms of Jaccard similarity and CorLoc.

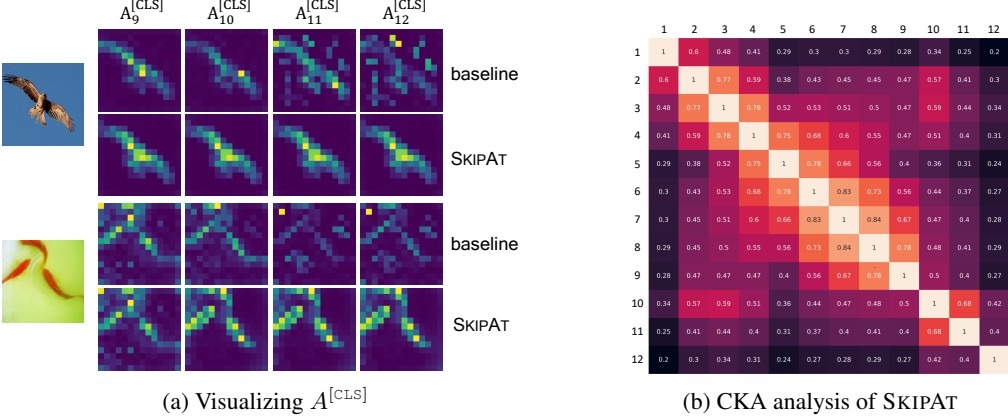

(a) Visualizing $A^{[\text{CLS}]}$        (b) CKA analysis of SKIPAT

Figure 5: (a) Mean of the attention of different heads from $A^{[\text{CLS}]}$ from last four layers of ViT-T/16 on the validation set of ImageNet-1K. Attention maps shows SKIPAT localizes the object better than vanilla ViT. (b) *CKA analysis* of SKIPAT shows that $Z^{\text{MSA}}$ has lower correlation between layers. The high correlation is between consecutive layers 2 through 8, where the MSA operation is skipped.

**Performance on mobile device.** To verify the efficiency of SKIPAT on low-power devices, we measure its inference time (averaged over 20 iterations) on a Samsung Galaxy S22 device powered by Qualcomm Snapdragon 8 Gen 1 Mobile Platform[*] with a Qualcomm Hexagon™ processor for image resolutions of $224 \times 224$ and $384 \times 384$ using ViT-T/16. The inference is performed on Neural Processing Unit in 8-bit precision. As shown in Table 2b, SKIPAT improves the runtime by 19% for image size of $224 \times 224$. The gain is even larger at 34% for image resolution $384 \times 384$, since the number of token increases. Thus, skipping computationally-heavy MSA blocks increases throughput by large margins and is confirmed even on mobile hardware.

**Semantic Segmentation on ADE20K** We show the performance of SKIPAT to dense prediction tasks such as semantic segmentation on ADE20K (Zhou et al., 2017). We follow (Liu et al., 2022; 2021) and use MMSegmentation (Contributors, 2020). We observe from Table 3, that SKIPAT outperforms all variants of ViT with 15% fewer FLOPs and 25% improved throughput. Interestingly, SKIPAT-S (ViT-S + SKIPAT) achieves 8% higher mIoU while being faster than ViT-T. Furthermore, SKIPAT-S has comparable mIoU with Swin-T (Liu et al., 2021) whilst having $3\times$ fewer FLOPs

---

[*]Snapdragon and Qualcomm branded products are products of Qualcomm Technologies, Inc. and/or its subsidiaries.

| METHOD | PSNR↑ | SSIM↑ | GFLOPs↓ | THROUGHPUT↑ |
|---|---|---|---|---|
| UNet (Ronneberger et al., 2015) | 39.65 | - | 35 | – |
| DAGL (Mou et al., 2021) | 38.94 | 0.953 | 255 | – |
| DeamNet (Ren et al., 2021) | 39.47 | 0.957 | 145 | – |
| MPRNet (Zamir et al., 2021) | 39.71 | 0.958 | 573 | – |
| NBNet (Cheng et al., 2021) | 39.75 | 0.959 | 91 | – |
| Restormer (Zamir et al., 2022) | 40.02 | 0.960 | 140 | – |
| Uformer-T (Wang et al., 2022b) | 39.66 | – | 12 | 17.6 |
| Uformer-T + SKIPAT | **39.69** | 0.959 | **11** | **22.2** |
| Uformer-S (Wang et al., 2022b) | 39.77 | 0.959 | 44 | 15.1 |
| Uformer-S + SKIPAT | **39.84** | **0.960** | **39** | **18.9** |
| Uformer-B (Wang et al., 2022b) | 39.89 | **0.960** | 89 | 9.2 |
| Uformer-B + SKIPAT | **39.94** | **0.960** | **77** | **10.9** |

Table 4: *Image denoising* on SIDD dataset using PSNR and SSIM (Wang et al., 2004) as the evaluation metrics in the RGB space. FLOPs and throughput are calculated on the input size of $256 \times 256$, on a single NVIDIA V100 GPU, averaged over the test set of SIDD.

and being $1.7\times$ faster. Comparing to fully convolution-based architectures, SKIPAT-T (ViT-T + SKIPAT) is on par with ResNet-18 in mIoU while having $4.7\times$ fewer FLOPs and being $1.8\times$ faster.

**Image Denoising** SKIPAT can also generalize to low-level tasks such as image denoising on SIDD (Abdelhamed et al., 2018b), which consists of images with real-world noise. We apply SKIPAT to Uformer (Wang et al., 2022b), a SoTA image denoising model, which is a U-shaped hierarchical network with Swin transformer blocks as the encoder and decoder. Detailed implementation of SKIPAT on Uformer is in the Appendix. Following the settings in (Wang et al., 2022b), we observe in Table 4 that SKIPAT outperforms the baseline Uformer variants with the 25% higher throughput on average. Furthermore, we observe that SKIPAT-B (Uformer-B + SKIPAT) achieves comparable performance with Restormer (Zamir et al., 2022), in terms of PSNR and SSIM, while having $2\times$ fewer FLOPs. Thus, we show the ability of SKIPAT to generalize to different tasks and also across architectures. Experiments on video denoising are provided in the Appendix.

## 4.2 ABLATIONS

All ablations are performed using ViT-T/16 on ImageNet-1K for 100 epochs to reduce the training time. Unless specified, following SKIPAT we skip the MSA blocks from layer 3 through 8 for all ablations. Additional ablations are provided in the supplementary material.

**Parametric function $\Phi$.** We study the effect of different parametric functions. As discussed in subsection 3.3, $\Phi$ can be as simple as an identity function, where we directly reuse representations from a previous MSA block into one of more subsequent MSA blocks. From Table 5, using an identity function results in a 4.7% drop in top-1 accuracy while being 47% faster than baseline ViT. Using

| FUNCTION $\Phi$ | KERNEL | CHANNEL EXPANSION | TOP-1↑ (%) | THROUGHPUT↑ (img/sec $\times 10^3$) |
|---|---|---|---|---|
| ViT-T | - | - | 65.8 | 5.8 |
| IDENTITY | - | - | 61.1 | 8.5 |
| CONV | $5 \times 5$ | - | 65.4 | 5.2 |
| DwC | $5 \times 5$ | - | 65.6 | 7.8 |
| | $3 \times 3$ | | 67.1 | 7.3 |
| SKIPAT | $5 \times 5$ | 2 | **67.7** | 6.9 |
| | $7 \times 7$ | | 67.4 | 6.6 |
| | | 0.5 | 64.4 | 7.4 |
| SKIPAT | $5 \times 5$ | 1 | 65.9 | 7.2 |
| | | 2 | **67.7** | 6.9 |

Table 5: *Ablations* using ViT-T/16 on ImageNet-1K for 100 epochs. We measure throughput (image/sec) with a batch size of 1024 on a single NVIDIA A100 GPU, averaged over the validation set of ImageNet-1K.

a convolution or DwC (Chollet, 2017) with kernel size $5 \times 5$ as a parametric function leads to the same performance as the baseline. However, DwC is 0.2% better and 50% faster than convolution, and 34% faster than the baseline. SKIPAT parametric function outperforms all.

**Kernel size.** By default SKIPAT uses a DwC with kernel size of $5 \times 5$. As shown in Table 5, using a $3 \times 3$ kernel is faster than default SKIPAT by 6%, but it is 0.6% worse in accuracy. A larger kernel size has poor accuracy and lower throughout. Irrespective of the kernel size, SKIPAT outperforms the baseline ViT-T by at least 1.4%, showing its ability to encode cross-token interactions.

**Channel expansion.** In the SKIPAT , the first linear layer $FC_1$, expands the channel dimension from $d \to 2d$. Table 5 shows the impact of channel dimension, *i.e.*, when the channel expansion ratio of $FC_1$ is 1.0 ($d \to d$) and 0.5 ($d \to d/2$). We observe that while the lower channel expansion ratio improves the throughput, it performs worse than default SKIPAT. This could be due to sub-optimal representations encoded by the DwC due to fewer filters.

## 5  CONCLUSION

We proposed SKIPAT, a plug-in module that can be used in any ViT architecture to reduce self-attention computations. SKIPAT leverages the dependency across MSA blocks and bypasses attention computation by re-using attention from previous MSA blocks. We introduced a simple and light parametric function that does not affect the inductive bias encoded in MSA. The SKIPAT function captures cross-token relations and outperforms the baseline while being computationally faster in terms of throughput and FLOPs. We plugged SKIPAT in different transformer architectures and showed its effectiveness on 7 different tasks.

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

# 6 IMPLEMENTATION DETAILS

## 6.1 HYPER-PARAMETERS

**ImageNet-1K: Image classification.** We train SKIPAT on the ILSVRC-2012 dataset (Deng et al., 2009) with 1000 classes (referred as ImageNet-1K). We follow the experimental settings of DeIT (Touvron et al., 2021) and use the codebase from the timm library (Wightman, 2019) to train ViT-T, ViT-S and ViT-B. We use the default $16 \times 16$ patch size, using an image resolution of $224 \times 224$ with total number of tokens $n = 196$. We train baseline ViT and SKIPAT for 300 epochs from scratch on 4 NVIDIA A100 GPUs using batch sizes of 2048 for ViT-T and 1024 for ViT-S and ViT-B.

**ImageNet-1K: Self-supervised learning.** We follow the experimental settings of DINO (Caron et al., 2021) and pre-train DINO and SKIPAT on ImageNet-1K using ViT-S/16 as the backbone. While likely the hyperparameters could be tuned further for our proposed SKIPAT method, we use same hyper-parameters for both the baseline and ours, yielding a conservative estimate of our model's performance. We pre-train both methods from scratch for 100 epochs using 4 NVIDIA A100 GPUs. For linear-probing, we freeze the backbone from the pre-training stage and fine-tune the classifier for 100 epochs, exactly as done in (Caron et al., 2021).

**Pascal-VOC2012: Unsupervised object segmentation.** We use the Pascal VOC 2012 (Everingham et al.) validation set for this experiment, containing 1449 images. We follow DINO and obtain unsupervised segmentation masks by thresholding the averaged self-attention map (extracted from the last layer of a pretrained ViT/SKIPAT model) to keep 80% of the mass. The Jaccard similarity $J$ between a predicted mask, $P$, and ground-truth mask, $G$, is defined as:

$$J(P, G) = \frac{G \cap P}{G \cup P}$$

We report Jaccard similarity, averaged over all the samples.

**ADE20K: Semantic segmentation.** We evaluate SKIPAT on ADE20K (Zhou et al., 2017), a widely-used semantic segmentation dataset, covering 150 semantic categories. The dataset includes 20K and 2K images in the training and validation set, respectively. Different variants of SKIPAT are evaluated using UperNet (Xiao et al., 2018) as the backbone. We use our ImageNet-1K pretrained model to initialize the backbone and Kaiming (He et al., 2015) initialization for other layers. We use AdamW (Loshchilov & Hutter, 2017), with an initial learning rate of $6e - 5$, weight decay of $1e - 2$, and linear warmup of 1500 iterations. All models are trained for $160K$ iterations with a batch size of 16 using MMSegmentation repo (Contributors, 2020). We keep the same hyper-parameters for SKIPAT and ViT.

**SIDD: Image denoising.** We follow the experimental settings in Uformer (Wang et al., 2022b) and train SKIPAT on the Smartphone Image Denoising Dataset (SIDD) (Abdelhamed et al., 2018a) which consists of real-world noise. The training samples are first randomly cropped to $128 \times 128$ patches and input to the model, which is trained for 250 epochs using batch size 32. The model is then evaluated on images of size $256 \times 256$.

**DAVIS: Video denoising.** We further apply our model to the temporal task of video denoising. We adopt the same U-shape encoder-decoder based architecture of UFormer. As the encoder and decoder backbone, we use UniFormer (Li et al., 2022). We train the model on noise level $\sigma = 30$ using Charbonnier loss (Charbonnier et al., 1994) on patches of $7 \times 128 \times 128$ using a multiple-input, multiple-output (MIMO) paradigm (Liang et al., 2022) (*i.e.*, the model outputs 7 reconstructed frames from 7 input frames). During inference, a video is divided into 3D patches of $7 \times 128 \times 128$ with an overlap of 10 pixels. Each patch is fed to the model and the outputs are merged to obtain the final denoised video. Following (Tassano et al., 2020), PSNR is calculated as averaged over videos. We use the same training hyper-parameters as image denoising.

## 6.2 ARCHITECTURE

**Image Classification.** All baseline ViT variants have 12 layers in total, which remains unchanged with SKIPAT. Following the CKA analysis of $Z^{\text{MSA}}$ in Figure 3(b) of our main paper, we skip computing the MSA blocks in layer 3 through 8 for all ViT variants and retrain it from scratch.

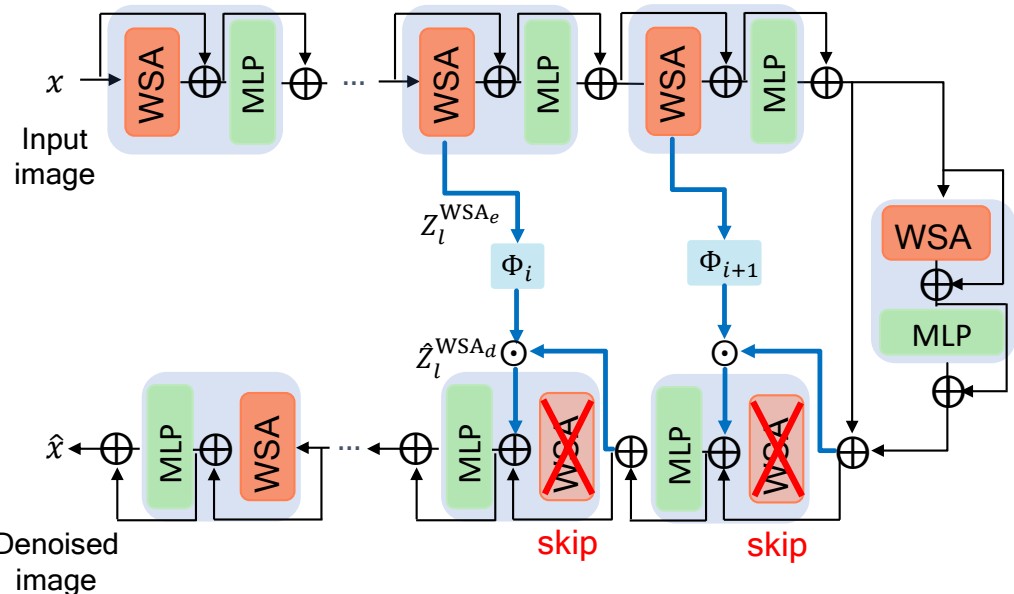

Figure 6: *Framework of* SKIPAT *on Uformer* Instead of standard MSA block in ViT, Uformer uses window self-attention (WSA) block similar to Swin Transformer. We skip WSA block in the layers close to the bottleneck.

**Image Denoising.** We apply SKIPAT to Uformer (Li et al., 2022) a SoTA image denoising model. Uformer is a U-shaped hierarchical network with Swin transformer blocks as the encoder and decoder, and skip connections between them. In SKIPAT, we skip window self-attention (WSA) block in each decoder block by reusing attention of the corresponding encoder block via SKIPAT parametric function. Let $Z_l^{\text{WSA}_e} \in \mathbb{R}^{n \times c}$ denote the output of the WSA block at layer $l$ from the encoder and $Z_{l-1}^d \in \mathbb{R}^{n \times c}$ denote the output of the layer $l-1$ from the decoder of Uformer. The input to the WSA block (which is skipped) at layer $l$ of the decoder is given by

$$\hat{Z}_l^{\text{WSA}_d} = \Phi(Z_l^{\text{WSA}_e}; Z_{l-1}^d) \in \mathbb{R}^{n \times 2c} \tag{9}$$

Here, ";" denotes concatenation along the channel dimension. We show the framework of SKIPAT on Uformer in Figure 6

### 6.3 VIDEO DENOISING

We apply our model to the temporal task of video denoising. As encoder and decoder backbone, we use UniFormer (Li et al., 2022), a U-shaped hybrid encoder-decoder architecture with 3D convolutions and spatio-temporal global self-attention blocks. The encoder of UniFormer comprises two 3D convolution layers followed by two spatio-temporal transformer layers with global self-attention (MSA) blocks. A downsampling operation is used after every layer in the encoder. The decoder is symmetric to the encoder with two transformer layers followed by two 3D convolution layers with an upsampling operation between each layer. Similar to Uformer, skip connections are used between encoder and decoder. Similar to image denoising, we skip MSA blocks in the decoder, however, simply adopt a naive SKIPAT, where we reuse window self-attention matrix, $A$, of the corresponding encoder block using an Identity function. Let $A_l^e \in \mathbb{R}^{n \times n}$ denote the self-attention matrix at layer $l$ from the encoder. The self-attention in the decoder stage at layer $l$ is given by $A_l^d = I(A_l^e) \in \mathbb{R}^{n \times n}$, where $I(.)$ is the identity function. We empirically observe that reusing attention works better in this task, and shows the ability of our method to be applied for different scenarios. We follow the experimental settings in (Tassano et al., 2020) and train SKIPAT on DAVIS (Pont-Tuset et al., 2017) dataset. We train using Charbonnier loss (Charbonnier et al., 1994) on patches of $7 \times 128 \times 128$ using a multiple-input, multiple-output (MIMO) paradigm (i.e. the model outputs 7 reconstructed frames from 7 input frames) for noise level $\sigma = 30$. From Table 6, we observe that SKIPAT performs on par

| METHOD | FastDVDNet (Tassano et al., 2020) | PaCNet (Vaksman et al., 2021) | VRT (Liang et al., 2022) | UniFormer (Li et al., 2022) | UniFormer+ SKIPAT |
|---|---|---|---|---|---|
| PSNR↑ | 34.04 | 34.79 | 36.52 | 35.24 | 35.16 |
| GFLOPS↓ | 41.9 | 34.8 | 708.8 | 93.2 | 77.1 |

Table 6: *Video denoising* Quantitative comparison (average RGB channel PSNR) with state-of-the-art methods for video denoising on DAVIS, with additive noise level $\sigma = 30$. FLOPs are calculated per frame per patch size of $256 \times 256$.

| BACKBONE | METHOD | TOP-1 (%) | PARAM ($\times 10^6$) | GFLOPS | THROUGHPUT (img/sec $\times 10^3$) |
|---|---|---|---|---|---|
| | T2T-ViT (Yuan et al., 2021) | 71.7 | 5.8 | 1.1 | – |
| | ConvNeXt (iso) (Liu et al., 2022) | 72.7 | 5.7 | 1.1 | 5.8 |
| | ViT (Dosovitskiy et al., 2020) | 72.8 | 5.7 | 1.2 | 5.8 |
| | A-ViT (Yin et al., 2022) | 71.0 | 5.7 | 0.8 | 6.3 |
| | Dynamic ViT (Rao et al., 2021) | 70.9 | – | 0.9 | 6.1 |
| ViT-T/16 | SViTE (Chen et al., 2021) | 71.7 | **4.0** | 0.9 | 6.2 |
| | SPViT (Kong et al., 2022) | 72.7 | 5.7 | 0.9 | 6.7 |
| | ATS (Fayyaz et al., 2022) | 72.7 | 5.7 | 0.9 | 6.1 |
| | PS-ViT (Tang et al., 2022) | 72.6 | – | **0.7** | 6.6 |
| | HVT (Pan et al., 2021) | 70.2 | 5.7 | **0.7** | **7.2** |
| | SKIPAT | **72.9** | 5.8 | 1.1 | 6.9 |
| | ConvNext-T (Liu et al., 2022) | **82.1** | 29.0 | 4.5 | 2.6 |
| | ConvNeXt (iso) (Liu et al., 2022) | 79.7 | 22.4 | 4.3 | 3.3 |
| | Swin-T (Liu et al., 2021) | 81.3 | 28.3 | 4.5 | 2.5 |
| | T2T-ViT (Yuan et al., 2021) | 80.7 | 21.5 | 5.2 | – |
| | CoaT-Lite-S (Xu et al., 2021) | 81.9 | 20 | 4.0 | – |
| | CoAtNet-0 (Dai et al., 2021) | 81.6 | 25 | 4.2 | – |
| | Poolformer-S24 (Yu et al., 2022) | 80.3 | 21.0 | 3.4 | – |
| | Twins-SVT-S (Chu et al., 2021) | 81.7 | 24.0 | 2.8 | – |
| | MobileViT-S (Mehta & Rastegari, 2021) | 78.4 | **5.6** | **2.0** | – |
| | PVT (Wang et al., 2021a) | 79.8 | 24.5 | 3.8 | – |
| ViT-S/16 | ViT (Dosovitskiy et al., 2020) | 79.8 | 22.4 | 4.6 | 3.2 |
| | A-ViT (Yin et al., 2022) | 78.6 | 22.4 | 3.6 | 3.4 |
| | Dynamic ViT (Rao et al., 2021) | 78.3 | 23.1 | 3.4 | 3.6 |
| | SViTE (Chen et al., 2021) | 80.2 | 13.1 | 2.7 | 3.5 |
| | ATS (Fayyaz et al., 2022) | 79.7 | 22.4 | 2.9 | 3.3 |
| | PS-ViT (Tang et al., 2022) | 79.4 | – | 2.6 | 3.9 |
| | SPViT (Kong et al., 2022) | 79.3 | 22.1 | 2.7 | 3.5 |
| | Rev-ViT (Mangalam et al., 2022) | 79.8 | 22.4 | 4.6 | 3.6 |
| | HVT(Pan et al., 2021) | 78.0 | 22.5 | 2.4 | **4.1** |
| | UniFormer-S (Li et al., 2022) | **82.9** | 3.6 | 1.8 | – |
| | EdgeViT-S (Pan et al., 2022a) | 81.0 | **1.9** | – | – |
| | SKIPAT | 80.2 | 22.1 | 4.0 | 3.8 |
| | Swin-S (Liu et al., 2021) | **83.5** | 88.0 | 15.4 | 1.0 |
| | Twins-SVT-B (Chu et al., 2021) | 83.2 | 56.0 | 8.6 | – |
| | PVT (Wang et al., 2021a) | 81.7 | 61.4 | **9.8** | – |
| | ConvNeXt (iso) (Liu et al., 2022) | 82.0 | 87.3 | 16.9 | 1.3 |
| ViT-B/16 | ViT (Dosovitskiy et al., 2020) | 81.8 | 87.3 | 17.6 | 1.2 |
| | SViTE (Chen et al., 2021) | 81.6 | **52.0** | 11.5 | 1.3 |
| | Rev-ViT (Mangalam et al., 2022) | 81.5 | 87.3 | 17.6 | 1.2 |
| | PS-ViT (Tang et al., 2022) | 81.5 | – | **9.8** | **1.6** |
| | SKIPAT | 82.2 | 86.7 | 15.2 | 1.5 |

Table 7: *Image classification on ImageNet-1K.* Accuracy *vs.* efficiency comparison of SKIPAT with SoTA methods for image resolution $224 \times 224$. For all the methods, we measure throughput (image/sec) with a batch size of 1024 on a single NVIDIA A100 GPU, averaged over the validation set of ImageNet-1K.

with baseline Uniformer, while having 17% fewer FLOPs. This shows that SKIPAT can generalize to temporal tasks.

# 7 ADDITIONAL EXPERIMENTS

**Image classification.** Here we extend our SoTA comparison with methods that go beyond vanilla ViT architectures. These methods include hierarchical (Swin, PVT, Poolformer, MobileViT, Twins-SVT) and Hybrid (ConvNext, CoAT) architectures. We provide the complete set of SoTA methods that improve the efficiency of ViT either by token sampling (extending Table 1 in our main paper), using hybrid architectures or window self-attention blocks in Table 7. Apart from methods that perform efficient token sampling, none of the other methods are directly comparable because they modify the underlying architecture of ViT, either by using window self-attention blocks or reducing the overall number of transformer layers.

**Self-Supervised Learning with DINO** Next, we show the generality of SKIPAT as its use in the backbone for self-supervised representation learning (SSL), using DINO (Caron et al., 2021). Since, SSL methods are quite expensive in the pretraining stage in terms of compute and training time, we illustrate that SKIPAT achieves comparable performance to using a ViT but with shorter training time. Following the experimental settings of DINO (Caron et al., 2021), we use ViT-S/16 (Dosovit-skiy et al., 2020) as our student and teacher networks with SKIPAT parametric function. We pretrain both baseline and ours using DINO for 100 epochs. We observe that SKIPAT achieves almost the same performance as fully trained DINO with around 26% less training time (73.3% in 96 GPU-hours *vs.* 73.6% in 131 GPU-hours). When trained on 100 epochs, we observe that SKIPAT outperforms DINO by 0.5% (74.1% *vs.* 73.6%).

**Unsupervised segmentation of DINO.** We follow DINO (Caron et al., 2021) and evaluate the performance of baseline DINO *vs.* SKIPAT on unsupervised object segmentation on Pascal-VOC2012 (Everingham et al.) dataset. We follow the experimental setting as discussed in section 6 and observe that baseline DINO has a Jaccard similarity of 45.3 while SKIPAT achieves 44.7. While SKIPAT outperforms DINO on image classification by 0.5%, we achieve comparable performance in terms of unsupervised object segmentation.

# 8 ADDITIONAL ABLATIONS

**Reusing self-attention.** As mentioned in Subsection 3.3, we skip the $Z^{\text{MSA}}$ in SKIPAT as the compute and memory benefit from skipping the entire MSA block is greater than skipping just the self-attention operation. Here we study the effect of skipping just the self-attention operation. Let $A_{l-1}$ denote the self-attention matrix at layer $l-1$, then the self-attention matrix at layer $l$ is given by $\hat{A}_l = I(A_{l-1})$. Similar to SKIPAT we skip computing the self-attention matrix from layers 3 through 8. As parametric function $\Phi$, we use an identity mapping and train ViT-T/16 from scratch for 100 epochs on ImageNet-1K. We observe from Table 9, that skipping the self-attention matrix results in a top-1 accuracy of 63.2% which is 2.1% higher than the skipping $Z^{\text{MSA}}$ with an identity function (61.1% - Table 7 of main paper). However, skipping self-attention matrix results in 20% decrease in throughput (8500 $\rightarrow$ 6800 images/sec) as compared to using an identity function to skip MSA block. It is interesting to note that skipping self-attention matrix results in a lower drop in performance as compared to skipping MSA block. However, applying a parametric function to skip self-attention can be challenging due to the properties of the self-attention matrix, and we leave this to future work.

**SKIPAT in pretrained model.** As mentioned in subsection 6.2, we train SKIPAT with all variants of ViT from scratch. For completeness, we also study the effect of skipping the self-attention matrix and the MSA block on a pretrained ViT-T using an Identity function, *without retraining*. We observe from Table 9 that skipping the self-attention computation in layers 3 through 8, results in a top-1 accuracy of 53.9%, while skipping MSA blocks results in top-1 accuracy of 47.8%. It is interesting to note that the drop in top-1 accuracy from skipping self-attention is merely 19% (72.8 $\rightarrow$ 53.9) on average and does not result in an *extremely* large drop as one might expect. This shows that there indeed exists high correlation across self-attention and $Z^{\text{MSA}}$, which SKIPAT utilizes to improve the efficiency of the ViTs.

**Skipping MSA in alternate configuration.** Instead of skipping the MSA operation in the layers $3-8$, we study the effect of skipping MSA operation at $l \in \{3,5,7,9\}, \{3,4,5,6\}, \{3,4,5,6,7,8,9,10\}$ instead of default $\{3,4,5,6,7,8\}$ in Table 8. We observe the default configuration outperforms the baseline ViT by 1.9% (65.8 *vs.* 67.7%) while being computationally faster in terms of throughput.

| METHOD | LAYERS | TOP-1 | THROUGHPUT (img/sec $\times 10^3$) |
|---|---|---|---|
| ViT-T | – | 65.8 | 5.8 |
| SKIPAT | {3,4,5,6} | 67.4 | 6.3 |
|  | {3,5,7,9} | 67.5 | 6.3 |
|  | {3,4,5,6,7,8,9,10} | 64.2 | **7.3** |
| SKIPAT (default) | {3,4,5,6,7,8} | **67.7** | 6.9 |

Table 8: *Ablations* using ViT-T/16 on ImageNet-1K for 100 epochs studying different layers that can be approximated.

| METHOD | TRAINING | TOP-1 (%) | THROUGHPUT |
|---|---|---|---|
| $A$ | ✓ | 63.2 | 6800 |
| $Z^{\text{MSA}}$ | ✓ | 61.1 | 8500 |
| $A$ | ✗ | 53.9 | 6800 |
| $Z^{\text{MSA}}$ | ✗ | 47.8 | 8500 |

Table 9: *Ablations* on the effect of skipping the self-attention, $A$, and the MSA block, $Z^{\text{MSA}}$. In the first two rows, models are trained for 100 epochs. In the last two rows we use a pretrained ViT-T/16 and simply skip computations in blocks 3-8 during inference. For all the experiments with use Identity function as $\Phi$.

## 9 CKA ANALYSIS OF ATTENTION FROM ViT-T

As discussed in Section 3.2 of our main paper, we analyze the CKA of the self-attention matrix for all tokens between different layers of ViT-T/16 pretrained on ImageNet-1K. Since in the supervised setting $A \in \mathbb{R}^{(n+1) \times (n+1)}$, we first remove the CLS token to obtain $A^P \in \mathbb{R}^{n \times n}$. We then compute the CKA of $A_l^P$ for $l \in \mathcal{L}$. We visualization the attention maps for two random patches in Figure 8. We observe similar correlation patterns as observed for CLS token. We akso observe from Figure 7, that there exists a high correlation across all the tokens from the self-attention matrix. Thus, reusing self-attention from different layers of the ViT can improve the overall throughput while yielding comparable accuracy as the baseline ViT.

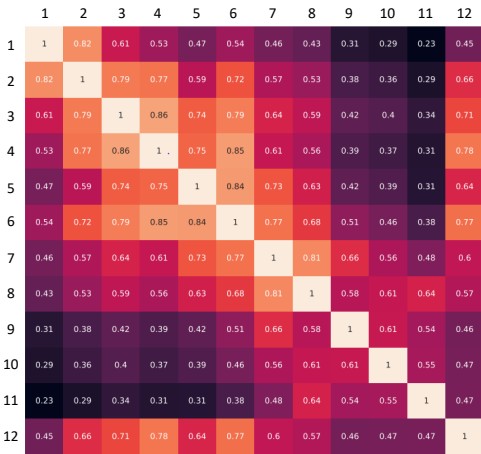

Figure 7: *CKA analysis of $A$* for all tokens from pretrained vanilla ViT-T/16 on the validation set of ImageNet-1K. We observe a high correlation for all tokens in $A$ from layers 1 to 8.

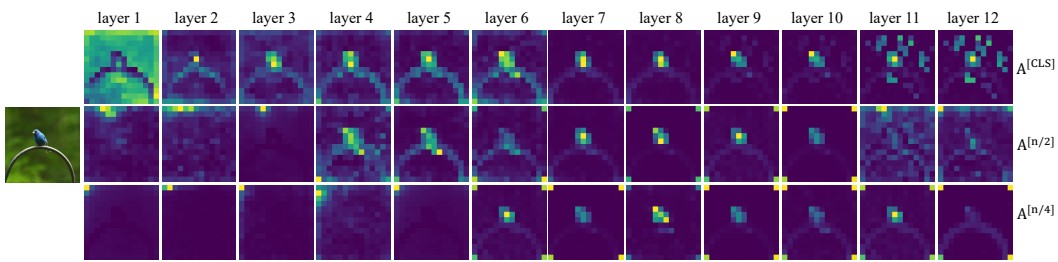

Figure 8: Mean of the attention heads from the CLS, $n/2^{\text{th}}$ and $n/4^{\text{th}}$ patch of a pretrained ViT-T/16 from the validation set of ImageNet-1K. $n = 196 = $ number of patches.

