# OpenReview forum: "Skip-Attention: Improving Vision Transformers by Paying Less Attention"
_ICLR.cc/2024/Conference — ICLR 2024 poster_

### Official Review · Reviewer_f6zn · 2023-10-28

**Soundness:** 2 fair
**Presentation:** 3 good
**Contribution:** 2 fair
**Rating:** 6
**Confidence:** 4

**Summary:**

This paper presents a new attention mechanism, named skip attention, aiming to reduce the computional cost of vision transformers. It is based on a simple observation that the attention maps of  adjacent  transformer blocks share similar patterns. Authors propose to reuse the attention maps of the current block in the next several ones by introducing a series lightweight operations, like linear transformations and efficient channel attention.

**Strengths:**

- This paper is well written. In the introduction section, the authors clearly explain the motivation of this paper, which is originally from the visualization of the attention maps of ViTs. The presentation is also clearly. It is easy for readers to follow the work.

- The results are good. When applied different versions of ViTs, the proposed method receives clear improvement over the baselines.

**Weaknesses:**

- It seems that the motivation of this paper has been mentioned in Zhou et al. (Refiner: Refining self-attention for vision transformers). They observe that reusing the attention maps in the next transformer block does not brings performance drop. The authors should more clearly explain the differences between this paper and the work mentioned above.

- The baselines used in this paper are not recently proposed. The results are already not state-of-the-art compared to recent works, like CMT (CVPR'2022). I would like to see how would the performance go when the proposed approach is applied to recent state-of-the-art ViT models as they mostly did not change the self-attention part.

- Many ViT models are based on window self-attention, which is original proposed in Swin Transformer (ICCV'2021). The authors have shown that the proposed method works well for the original self-attention. So, how would the performance go when the proposed method is applied to ViTs with window self-attention.

- In my view, one of the important functionalities of this paper is to compress vision transformers. Maybe the authors can show more comparisons with methods for compressing ViTs, like DynamicViT and EViT. As I found the proposed approach can improve the baselines' performance with even less computations. This may better highlight the strength of this paper.

**Questions:**

I care more about the novelty of this paper as the originality of this paper has been mentioned in a previous work. If the authors make this clear, I would like to raise the ranking score.

---

> ### Author Response · Authors · 2023-11-18
> **Response to Reviewer f6zn**
>
> We appreciate the Reviewer f6zn's valuable feedback. We address the concerns as follows:
>
> **1. Comparison with Refiner**
>
> Thank you for the pointer. We discuss Refiner (Zhou et al.) in Section 2 under ``Efficient attention" (6th line from the end of the paragraph).
>
> We acknowledge that both SkipAT and Refiner leverage the high correlation across layers of a transformer, but for two different purpose: increasing the classification accuracy (by Refiner) and increasing the computational efficiency (by SkipAT). Refiner introduces additional convolutions on top of ViT to de-correlate the attention maps to increase the expressive power of the model. This obviously comes at an additional computational cost both in terms of parameters and latency compared to the vanilla ViT. In contrast, SkipAt removes the correlated MSA blocks and approximate them using a cheap convolutional alternative leading to a reduced computation compared to the vanilla ViT.
>
> On ImageNet-1K, Refiner achieves a top-1 accuracy of 80.3\% using 2M more parameters than SkipAT, which achieves a similar accuracy of 80.2\%. We shall clarify and further emphasize the novelty of SkipAT with respect to Refiner in the paper.
>
> **2. Comparison with CMT**
>
> We thank the reviewer for providing the reference. In Table 2(a), we compare SkipAT with methods that focus on improving the efficiency of transformers without changing the backbone. We also provide a comprehensive list of SoTA methods in Table 7 in the Appendix. We shall add CMT to Table 7.
>
> In Table 2(a), we apply SkipAT to SoTA transformer backbones such as PvT and LIT, and show that SkipAT is agnostic to transformer architecture and improves both top-1 accuracy and throughput of these backbones.
>
> **3. Use of SkipAT with window self-attention**
>
> In Section 4.1 -- ``Image Denoising" and Table 4, we describe the application of SkipAT to Uformer, which is a U-shaped hierarchical network with Swin transformer blocks as the encoder and decoder. We skip the whole window self-attention (WSA) block in each decoder block by reusing attention of the corresponding encoder block via the SkipAT parametric function. A detailed description is provided in Section 6.2 in Appendix.
>
> We observe from Table 4, that  SkipAT outperforms the baseline Uformer variants with the 25\% higher
> throughput on average.
>
> **4. Comparison with DynamicViT and EViT**
>
> We compare SkipAT with DynamicViT in Table 7 in the Appendix. When using ViT-T/16 as the backbone, SkipAT outperforms DynamicViT by 2\% in terms of top-1 accuracy and also achieves higher throughout (6900 vs 6100 im/sec).
>
> Comparing EViT [1*] with SkipAT using ViT-S/16 as backbone, SkipAT has comparable top-1 accuracy (80.2\% of SkipAT vs. 79.8\% of EViT) and comparable throughput (3800 vs/ 3900 im/sec). We shall add these results in Table 7.
>
> [1*] Liang *et al.*, Not All Patches are What You Need: Expediting Vision Transformers via Token Reorganizations, ICLR 2022

---

> > ### Comment · Reviewer_f6zn · 2023-11-20
> >
> > Thanks for the feedback. The authors have solved most of my concerns. I'd like to lift the score to 6.

---

> > > ### Author Response · Authors · 2023-11-21
> > > **Thank you R-f6zn**
> > >
> > > We sincerely appreciate R-f6zn's time and effort in evaluating our work, for the insightful comments, and for raising the score to "6". We shall add the comparisons with CMT, EViT and also explain the differences with Refiner in the camera ready version of the paper.

---

### Official Review · Reviewer_iYVP · 2023-10-29

**Soundness:** 3 good
**Presentation:** 3 good
**Contribution:** 3 good
**Rating:** 8
**Confidence:** 4

**Summary:**

This paper improves the efficiency of Vision Transformer by replacing some attention layers with a compute-efficient parametric function, ie, convolutional feed-forward layer. The idea is motivated by a clear observation and analysis that attention patterns tend to be redundant between different layers, indicating a strong correlation. With the novel design, the authors validated the framework on various architecture and datasets. Comprehensive experiments have shown the advantage of their method.

**Strengths:**

1. The motivation of this paper is very clear, accompanied by strong analysis in the attention patterns.
2. The figures and visualizations can clearly demonstrate their method. The overall presentation is good to me.
3. Experiments are comprehensive, including different architectures, datasets, tasks, which strongly demonstrate that the proposed method is general.
4. The performance gain is also consistent across different settings.

**Weaknesses:**

1. Based on the analysis in Section 3.2, it makes sense for the authors to apply their method from layer 2 to 8. However, it is not convincing for different pretrained ViTs to skip layer 2 to 8 as well if considering different training objectives or pretrained datasets. Thus, it would be better for the authors to study if other pretrained ViTs (MAE [A], DINOv2 [B], SAM [C]), have the same phenomenon.

2. Introducing convolution into ViTs has shown to be effective in related works [D], which is intuitive to me to achieve performance gain for SKIPAT. In this paper, SKIPAT adopts convFFN as a parametric function to replace MSAs, which still needs to be trained from scratch in order to achieve efficiency gain. It would be promising if this parametric function can be used as a drop-in replacement for existing large ViTs.


[A] He, Kaiming, et al. "Masked autoencoders are scalable vision learners." Proceedings of the IEEE/CVF conference on computer vision and pattern recognition. 2022.

[B] Oquab, Maxime, et al. "Dinov2: Learning robust visual features without supervision." arXiv preprint arXiv:2304.07193 (2023).

[C] Kirillov, Alexander, et al. "Segment anything." ICCV (2023).

[D] Wang, Wenhai, et al. "Pvt v2: Improved baselines with pyramid vision transformer." Computational Visual Media 8.3 (2022): 415-424.

**Questions:**

Can the authors specify more on the experimental setting of applying SKIPAT into hierarchical ViTs? I can understand that SKIPAT works for layer 2 to 8 in plain ViTs. But it is not intuitive to me how to select the layers to skip in PVT, LIT, etc.

---

> ### Author Response · Authors · 2023-11-18
> **Response to Reviewer iYVP**
>
> We appreciate the Reviewer iYVP's valuable feedback. We address the concerns as follows:
>
> **1. Different pretrained ViTs**
>
> Thanks for raising this interesting question: To what extent the correlation analysis from a supervised image classification model, as in Section 3.2, generalize to other tasks and datasets?
> We empirically observe that the optimal configurations of parametric function is determined mostly by the transformer architecture rather than the task.
>
>  When considering different training objectives, we applied SkipAT to DINO in Section 7 of the Appendix under ``Self-Supervised Learning with DINO". We observe that SkipAT achieves almost the same performance as fully trained DINO with around 26\% less training time (73.3\% in 96 GPUhours vs. 73.6\% in 131 GPU-hours).
> Moreover, we also use SkipAT on semantic segmentation on ADE20K (Table 3) and Image Denoising on SIDD (Table 4), showing SkipAT is generalizable to different training objectives.
>
> For other architectures, a principled way could be to compute the correlation matrix by computing the CKA between different layers and using a threshold to identify the layers on which SkipAT can be applied, as we have done for PvT and LIT on Table 2(a).
>
> **2. Using SkipAT in pretrained models**
>
> Indeed finding a parametric function, which can be plugged into any pretrained model is practically coveted but is very challenging given the complex interplay between the layers of a deep neural network.
> Surprisingly, our experiments demonstrate that using an identity function on a pretrained ViT-T/16 in a plug-and-play fashion could still perform reasonably well: only 9.3\% drop in performance, when copying self-attention matrix $A$ in layers 3-8 (Table 9 of the Appendix).
>
> Regularizing transformer blocks to be more correlated during training could be an step toward more robust plug-and-play adaptors.
>
> **3. Experimental setting in hierarchical ViTs**
>
> *Using SkipAT in PvT*
>
> As reported in Table 2, we applied SkipAT on a hierarchical architecture PvT-S, which consist of 4 stages and each stage consists of 3, 3, 6 and 3 transformer blocks respectively. We observe high correlation of $Z^{MSA}$ in the third stage and apply SkipAT in the intermediate blocks i.e. between 2 through 5 blocks. In this case, we still have applied SkipAT to adapt the features of the same dimensionality. However, by increasing the stride of the depth-wise separable convolutions in the parametric function, SkipAT can adapt features across different resolutions.
>
> *Using SkipAT in LIT*
>
> As reported in Table 2, we applied SkipAT in LIT-T and LIT-S, which consist of 4 stages. The first two stages consists of convolution layers and the last two stages consists of transformer blocks. We apply SkipAT in the intermediate transformer blocks in the third stage of LIT i.e. we skip layers 2 to 5 in the third stage.
>
> We will clarify this in the text.

---

> > ### Comment · Reviewer_iYVP · 2023-11-20
> > **Additional Comments**
> >
> > Thanks for the authors's rebuttal. I'm satisfied with the response. The additional experiments with other pretrained ViTs are interesting. It would be great to include these results in the appendix.

---

> > > ### Author Response · Authors · 2023-11-21
> > > **Thank you R-iYVP**
> > >
> > > We sincerely appreciate R-iYVP's time and effort in evaluating our work and for the insightful comments. We shall add the additional experiments on pretrained ViT to the Appendix in the camera ready version of the paper

---

### Official Review · Reviewer_VJcB · 2023-10-30

**Soundness:** 2 fair
**Presentation:** 3 good
**Contribution:** 3 good
**Rating:** 6
**Confidence:** 4

**Summary:**

A core component of the vision transformer is the self-attention layer, which is quadratic in the number of tokens. Following similar insights in (Raghu et al. 2022), the authors observe that self-attention operation is redundant at least in the intermediate layers, i.e. there is high correlation between:
* (CLS -> token) Attention maps between  at layer $L$ and layer $L - 1$.
* MSA representations between layer $L$ and layer $L - 1$.

Leveraging this insight, the authors propose to replace the more computationally intensive attention operation with a lightweight refinement module termed SkipAt. More specifically, MSA at depth $L$ is replaced with inverted bottleneck layers (depthwise convolutions sandwiched between two dense layers). SkipAt layer $L$ takes the output of SkipAt layer $L - 1$ as input, as opposed to Multi-Head Self-Attention (MSA) at layer $L$ which takes the output of the MLP layer $L - 1$ as input.

The authors show experiments on classification, segmentation, unsupervised object discovery and image (+video) denoising.  They plugin their SkipAt technique and attain improved throughput and in most cases, improved accuracy.

**Strengths:**

* The approach is simple and effective.
* The authors have tested their SkipAt approach on a number of tasks. Their approach improves over similar Vision Transformer backbones and leads to improved throughputs.
* The writing is crisp and clear.

**Weaknesses:**

Some experiments can be added which decouple the improvements obtained with convolutions vs the SkipAt formulation. I initially rate this  above bordeline. If the authors can convincingly answer my questions, I am happy to increase the score.

**Questions:**

## Major Requests:
-----------

* The main motivation of SkipAt is that the output representations of MSA in vision transformers are redundant. So, the paper claims that just refining the outputs of the previous MSA layers is sufficient. However, SkipAt consists of depthwise convolutions which are also quite powerful modules themselves, so it is unclear if the throughput gains come just by the convolutions rather than the SkipAt formulation. I suggest that the authors run the couple of ablations below, If these ablations reach lower accuracy, it would be convincing that the SkipAt formulation is responsible for the accuracy gains.
  * Replace all layers with SkipAt instead of just layers from 3 through 8. According to the authors hypothesis, since layers 9 though 12 have lesser correlation, using SkipAt at these layers should hurt accuracy.
  * To show the importance of skipping the attention blocks, in Eq 7) the authors can replace $\phi(Z_{l-1}^{MSA})$ with just $\phi(Z_{l-i})$. This will give more evidence that skipping the attention block is necessary.

* The authors test their module on Ti, B and S which all have 12 layers so they recommend to use SkipAt layers from 3 through 8. How do they recommend tuning these for larger depths?
* In page 6, authors say that $n >> d$ and so $O(n^d)$ term dominates. I would suggest the authors add that this is specific to dense prediction tasks, since for image classification even for a S model (d=384, and n=196), so the claim that n >> d is not general.
* Are the throughput increase in Fig a) significant? What do the error bars look like?

## Minor Comments:
-----------
These are just nice to have and are not likely to influence my final rating.

* The authors use the efficient channel module. But, the ablation is missing from Table 5.
* The figures from Raghu et al, indicate that there might be redundancy across the MLP layers as well. Does it make sense to have a Skip-MLP module?

## Minor suggestions:
------

Some suggestions to improve presentations:

* The authors can consider making the numbers in Figure 3 and Figure 5 bigger.
* It is clear by comparing Fig 3 b) and Fig 5 b), that Fig 5 b) has lower correlation. The authors can also have a line plot where the x-axis is the layer id and y-axis is the average correlation of layers before it. If we plot the baseline ViT and the ViT with SkipAt on the same graph, it will make the comaprison even clearer.
* In Fig 1), are the circles to indicate #params necessary since they are roughly the same? It gives the impression that the improvements are not significant even if they are, since the circles overlap

---

> ### Author Response · Authors · 2023-11-18
> **Response to Reviewer VJcB**
>
> We appreciate the Reviewer VJcB's valuable feedback. We address the concerns as follows:
>
> **1. Additional ablations**
>
> Thank you for this interesting suggestion. Based on the reviewer's suggestion, we implemented the SkipAT parametric function across all layers of VIT-T/16 and train for 100 epochs on ImageNet-1K. The default configuration, which employs SkipAT at layers 3 to 8, yields a top-1 accuracy of 67.7\% (Table 5 in the main paper). When SkipAT is applied to layers 1 through 12, the top-1 accuracy decreases to 54.1\%, indicating a significant drop of 13.6\%. This suggests that leveraging the parametric function is only beneficial for approximating MSA blocks with high correlation.
>
> Moreover, we observe that using an Identity function, where we simply copying $Z^{MSA}$ features across 3 to 8 layers, achieves decently high top-1 accuracy of 61.1\% (Table 5), witnessing high correlation in MSA blocks. Thus, we hypothesize that the performance gains is due to the SkipAT formulation. Additionally, using more powerful parametric function such as convolution layer marginally increases performance (top-1 accuracy: 65.4\%) but increases throughput by 39\% (8500 vs. 5200 im/sec) (Table 5).
>
> We understand the reviewer asks us to replace SkipAT parametric function $\phi(Z^{MSA}_{l-1})$ with an Identity function (i.e. using features from prev. blocks). If so, we mentioned the effect of using an Identity function above.
>
> We are happy to follow-up with the reviewer for any clarifications or additional ablations.
>
> **2. Using larger models**
>
> The reviewer raises an interesting point.
>
> For large models, a principled approach would be to compute the correlation matrix using the CKA analysis of $Z^{MSA}$ for every layer. Using a threshold on this correlation matrix would indicate the layers with high correlation, on which the SkipAT parametric function can be applied.
> We use this approach to apply SKipAT on pyramid (PvT) and hybrid (LIT) transformer architectures (Table 2(a)) where arrangement of transformer blocks are different than ViT. For example, PvT-S consists of 4 stages with 3, 3, 6 and 3 transformer blocks at each stage. We observe high correlation of $Z^{MSA}$ in the third stage and apply SkipAT in the intermediate blocks i.e. between 2 through 5 blocks.
>
> Another approach could be using more advanced techniques such as Neural Architectural Search (NAS). While NAS might propose valid solutions, it is computationally more expensive.
>
> **3. Complexity for dense prediction tasks**
>
> Thank you for the precise point: Indeed the claim holds for dense prediction tasks. We shall make this change.
>
> **4. Error bars of throughput**
>
> We report the mean and standard deviation of the throughput of SkipAT for 10 runs on ImageNet-1K using ViT-T/16, ViT-S/16 and ViT-B/16. For a single run, we measure throughput (image/sec $\times 10^3$) with a batch size of 1024 on a single NVIDIA A100 GPU, averaged over the validation set of ImageNet-1K. The results are in the table below:
>
> | Arch               | Throughput (im/sec $\times 10^3$) |
> | ------------------ | ---------------------------------- |
> | ViT-T/16           | 5.9 $\pm$ 0.3                     |
> | SkipAT + ViT-T/16  | 6.8 $\pm$ 0.2                     |
> | ViT-S/16           | 3.1 $\pm$ 0.3                     |
> | SkipAT + ViT-S/16  | 3.7 $\pm$ 0.3                     |
> | ViT-B/16           | 1.2 $\pm$ 0.2                     |
> | SkipAT + ViT-B/16  | 1.6 $\pm$ 0.1                     |
>
> **5. Minor comments and suggestions**
>
> Thank you very much for these insightful comments and suggestions. We shall incorporate all of them in the camera-ready version.

---

> > ### Comment · Reviewer_VJcB · 2023-11-22
> > **Rebuttal Response**
> >
> > Thanks for the additional experiments:
> >
> > All my comments are addressed except this:
> >
> > "To show the importance of skipping the attention blocks, in Eq 7) the authors can replace $\phi(Z_{l-1}^{MSA})$ with just  $\phi(Z_{l-i})$." The authors show what happens if $\phi$ is replaced by an identity function..
> > However the requested ablation was to simply replace the self-attention blocks in the original ViT formulation with depthwise convolutions. According to the mathematical notation, precisely, the request was for:
> >
> > * $\phi$ to be the same (i.e, depthwise convolution)
> > * the ablated input to $\phi$ to be $Z_{l-i}$ (See: Eq 4 and Eq 5) instead of $Z_{l-1}^{MSA}$.
> >
> > In particular, the authors claim still holds that they are able to outperform baseline transformers with the skip attention modules. But it is still unclear to me if the gains observed by the authors can come with just replacing self-attention in the baseline VisionTransformer with depthwise convolutions.

---

> > > ### Author Response · Authors · 2023-11-22
> > > **Response to Reviewer VJcB**
> > >
> > > Thank you very much for the clarification. We will run the experiment as suggested. We are afraid that we might not have the result before the rebuttal deadline as it would take us 3 days for training ViT-T/16. We will make sure to add this experiment to the camera-ready version of the paper.

---

### Meta-Review · Area_Chair_d5y6 · 2023-12-04

**Metareview:**

The authors identify redundancy in the self-attention operation of ViT models, particularly in intermediate layers. They observe a high correlation between:

Attention maps from the CLS (class) token to other tokens between adjacent layers, and Multi-Head Self-Attention (MSA) representations between adjacent layers.

Based on this observation, the authors propose a more computationally efficient alternative called SkipAt, a lightweight refinement module. In their experiments, the authors assess the impact of their SkipAt technique on tasks such as classification, segmentation, unsupervised object discovery, and image (plus video) denoising. By incorporating the SkipAt module, they achieve enhanced throughput and, in most instances, improved accuracy.

**Justification For Why Not Higher Score:**

- The proposed method is kind of ad hoc. One has to train the models and they identify the redundancy patterns. Then replace the attention layers with the proposed module. This is similar to model compression. The value to the community may be limited.

- The finding of the redundant attention patterns across different layers is not new.

**Justification For Why Not Lower Score:**

- The authors conducted comprehensive experiments in the submission and rebuttal.

- The performance gain is significant, demonstrating effectiveness of the proposed method.

- The idea is still interesting for inspiring the community to further investigate behavior of the ViT models.

---

### Decision · Program_Chairs · 2024-01-16

Accept (poster)